# Decorin Promotes Osteoblastic Differentiation of Human Periodontal Ligament Stem Cells

**DOI:** 10.3390/molecules27238224

**Published:** 2022-11-25

**Authors:** Orie Adachi, Hideki Sugii, Tomohiro Itoyama, Shoko Fujino, Hiroshi Kaneko, Atsushi Tomokiyo, Sayuri Hamano, Daigaku Hasegawa, Junko Obata, Shinichiro Yoshida, Masataka Kadowaki, Risa Sugiura, Mhd Safwan Albougha, Hidefumi Maeda

**Affiliations:** 1Department of Endodontology and Operative Dentistry, Division of Oral Rehabilitation, Faculty of Dental Science, Kyushu University, Fukuoka 812-8582, Japan; 2Department of Endodontology, Kyushu University Hospital, Fukuoka 812-8582, Japan; 3OBT Research Center, Faculty of Dental Science, Kyushu University, Fukuoka 812-8582, Japan

**Keywords:** decorin, periodontal ligament stem cells, osteoblastic differentiation, periodontal tissue regeneration

## Abstract

The aim of this study is to clarify the biological functions of decorin (DCN) in the healing and regeneration of wounded periodontal tissue. We investigated the expression pattern of DCN during the healing of wounded periodontal tissue in rats by immunohistochemistry and the effects of DCN on the osteoblastic differentiation of human periodontal ligament (PDL) stem cells (HPDLSCs) and preosteoblasts by Alizarin red S staining, quantitative reverse transcription-polymerase chain reactions, and western blotting. The expression of DCN was increased around the wounded PDL tissue on day 5 after surgery compared with the nonwounded PDL tissue, whereas its expression was not changed in the osteoblastic layer around the wounded alveolar bone. Furthermore, DCN promoted the osteoblastic differentiation of HPDLSCs, but it did not affect the osteoblastic differentiation of preosteoblasts. ERK1/2 phosphorylation was upregulated during the DCN-induced osteoblastic differentiation of HPDLSCs. DCN did not affect proliferation, migration, or the PDL-related gene expression of HPDLSCs. In conclusion, this study demonstrates that DCN has a role in the healing of wounded periodontal tissue. Furthermore, DCN secreted from PDL cells may contribute to bone healing by upregulating osteoblastic differentiation through ERK1/2 signaling in HPDLSCs, indicating a therapeutic effect of DCN in periodontal tissue regeneration.

## 1. Introduction

Oral health is an important factor for maintaining a good quality of life [1]. Teeth are essential for mastication, and tooth loss causes serious health problems with speech, chewing, and emotional and aesthetic factors [2].

Periodontal tissue consists of periodontal ligament (PDL) tissue, the alveolar bone, gingiva, and cementum. The alveolar bone is the mineralized tissue supporting the tooth, and PDL tissue is the fibrous connective soft tissue that surrounds the tooth root and connects it with the alveolar bone [3]. Furthermore, PDL tissue plays an essential role in the homeostasis, repair, and nutrition of the tooth [4]. Once periodontal tissue is defected by severe caries, periodontitis, and trauma, it is difficult to regenerate the lost tissue, and tooth loss can occur. Regeneration therapy of periodontal tissue was developed using mesenchymal stem cells (MSCs) and induced pluripotent stem cell-derived MSCs. However, these methods have several issues including ethical problems and difficulties in the management of running costs [5]. Therefore, to overcome these issues, it is necessary to identify the factors that promote the healing and regeneration of periodontal tissue.

PDL tissue contains PDL stem cells (PDLSCs), which have multipotency to differentiate into neural/glial cells, osteoblasts, cementoblasts, and adipocytes in vitro [6,7,8]. Furthermore, transplanted PDLSCs form cementum-/bone-like and PDL-like structures in vivo [7,9]. These reports indicate that PDLSCs are an important cell source for periodontal tissue healing and regeneration.

Decorin (DCN), which is a member of the small leucine-rich proteoglycan family, is a component of the extracellular matrix and has a core protein of approximately 40 kDa and a chondroitin sulfate/dermatan sulfate glycosaminoglycan chain. DCN is expressed in various tissues such as skin, bone [10], cornea [11], and aorta [12], and is an important molecule in wound healing [13]. *DCN*-null mice have an abnormal collagen structure and exhibit impaired healing in wounded skin tissue [14], whereas DCN treatment promotes wound healing of mouse skin by regulating collagen reorganization [15]. DCN has a crucial role in regulating the expression of proinflammatory cytokines [16]. Moreover, DCN production is upregulated by treatment with proinflammatory cytokines such as interleukin-1β (IL-1β) and tumor necrosis factor-α (TNF-α) [17,18]. DCN also plays a significant role in osteogenesis [19]. Osteoprogenitor cells produce DCN during the healing of rat tooth extraction sockets [20], and DCN promotes the proliferation and mineralization of osteoblasts in vitro [21].

The therapeutic effects of DCN have previously been investigated. In corneal wound healing, the sustained delivery of DCN through eye drops to the wounded cornea led to improved ocular function [22]. Furthermore, DCN is also used as the anti-scarring therapeutic factor in the healing of wounded spinal cords [23] and osteomyelitis [24]. Other reports demonstrated that the injection of DCN into a limb bone fracture site promoted bone healing in vivo [25].

DCN is expressed in periodontal tissue [26,27,28]. It is also reported to be expressed in PDL fibroblasts [29] and contributes to the development and maintenance of PDL tissue [30,31]. *DCN*-deficient mice have an abnormal morphology and organization of collagen fibrils in PDL tissue [31]. However, the effects of DCN on the healing of wounded periodontal tissue and the biological functions of DCN in human PDLSCs (HPDLSCs) have not yet been clarified. Therefore, we investigated the expression pattern of DCN in wounded rat periodontal tissue as well as the *DCN* gene expression in inflammatory cytokine-treated human PDL cells and preosteoblasts and explored the effects of DCN on the differentiation, proliferation, and migration of HPDLSCs and preosteoblasts.

## 2. Results

### 2.1. Expression of DCN in Wounded Rat Periodontal Tissue

To analyze the expression pattern of DCN during the healing of wounded periodontal tissue, immunohistochemical analysis was performed using a wounded periodontal tissue rat model. The localization of anti-DCN antibody-positive areas was examined in the wounded site of left maxillary first molars on days 1, 3, 5, 7, 14, and 28 after surgery. Preoperative specimens were used as the control (Figure 1A). The quantification of anti-DCN antibody-positive areas was performed in PDL tissue (Figure 1I) and the osteoblastic layer (Figure 1J) around the wounded site. The osteoblastic layer was determined by analyzing the expression of OSX, which is the marker of osteoblasts (Appendix A). On day 5 after surgery, more anti-DCN antibody-positive areas were identified around the wounded PDL tissue than in the normal PDL tissue (Figure 1D,I), whereas anti-DCN antibody-positive areas were not changed in the osteoblastic layers around the alveolar bone (Figure 1D,J). On days 1, 3, 7, 14, and 28 after surgery, no significant differences in the anti-DCN antibody-positive areas were found between the wounded and nonwounded sites (Figure 1B,C,E–G,I,J). Negative controls using a rabbit control IgG showed no positive staining (Figure 1H).

### 2.2. Expression of DCN in Human PDL Cells Treated with IL-1β and TNF-α

Our previous studies showed that the IL-1β expression increases in wounded rat periodontal tissue, including PDL tissue [32,33]. These results led us to examine the effects of IL-1β and TNF-α on the *DCN* expression in human PDL cells by quantitative RT-PCR (Appendix A). Consistent with previous reports [34], the *DCN* expression was significantly increased in IL-1β- and TNF-α-treated human PDL cells compared with that in untreated cells.

### 2.3. Effects of DCN on the Osteoblastic Differentiation of HPDLSCs and Preosteoblasts

We analyzed the effects of DCN on the osteoblastic differentiation of HPDLSCs. Alizarin red S-positive areas of HPDLSCs were remarkably increased in the DM + 2DCN group compared with those in the DM group (Figure 2A,B). Additionally, quantitative RT-PCR analysis and western blotting analysis showed that the expression of bone-related genes, such as *BMP2*, *BSP*, *OCN*, *OPN,* and *OSX,* and the protein expression of OPN were significantly increased in the DM + 2DCN group (Figure 2C–I).

We also analyzed the effect of DCN on the osteoblastic differentiation of Saos2 cells. Alizarin red S-positive areas of Saos2 cells were significantly increased in the DM + 0.5 DCN group compared with those in the DM group, but they were not comparable to those of HPDLSCs (Figure 3A,B). Furthermore, the quantitative RT-PCR analysis and western blotting analysis showed that the gene and protein expression of the bone-related markers were not increased after culturing with DCN (Figure 3C–H).

### 2.4. Phosphorylation of ERK1/2 in DCN-Treated HPDLSCs

DCN signaling is mediated through the ERK and AKT pathways in breast cancer cells [35], endothelial cells [36], and myoblasts [37]. Therefore, to investigate the intracellular signaling molecules involved in the DCN-induced osteoblastic differentiation of HPDLSCs, we examined the effects of DCN on the phosphorylation of proteins such as ERK1/2 and AKT in HPDLSCs. Western blotting showed that the p-ERK1/2 expression was significantly increased in the DM + 2DCN group compared with that in the DM group (Figure 4A,B). However, the p-AKT expression was not altered regardless of the DCN coating (Figure 4C,D).

### 2.5. Effects of DCN on the Proliferation, Migration, and PDL-Related Gene Expression of HPDLSCs

The effects of DCN on the proliferation and migration of HPDLSCs were examined using WST-1 proliferation and ring cell migration assays, respectively. The results showed that DCN did not affect the proliferation or migration of HPDLSCs (Figure 5). Furthermore, quantitative RT-PCR analysis showed that the expression of PDL-related genes, *FBN1*, *OPG*, *PLAP1*, and *POSTN,* was not increased in HPDLSCs treated with DCN compared with that in the control (Figure 6).

## 3. Discussion

This study reveals the expression pattern of DCN during the healing of wounded periodontal tissue and the effects of DCN on the osteoblastic differentiation of HPDLSCs and preosteoblasts. The expression of DCN was increased around wounded PDL tissue in the early stages of tissue healing, especially 5 days after surgery. Additionally, DCN promoted the osteoblastic differentiation of HPDLSCs but did not affect that of preosteoblasts. This is the first report showing that DCN may be an effective molecule to heal wounded periodontal tissue and induce the osteoblastic differentiation of HPDLSCs.

The healing of wounded periodontal tissue occurs through three stages: inflammation, tissue formation, and tissue remodeling [38]. The inflammation stage typically occurs over the first several days of tissue healing and finishes after 1 week [39,40]. During the inflammation stage, the expression of inflammatory cytokines such as IL-1β and TNF-α is upregulated. Previous reports have shown that IL-1β and TNF-α promote DCN production in human endometrial cells [17], immune cells [18], and human PDL cells [34]. Additionally, DCN controls inflammation by regulating the production of proinflammatory cytokines [16]. Using our animal model, we previously reported an increased expression of IL-1β around wounded PDL tissue and the alveolar bone from days 1 to 3 after surgery [33] and the emergence of Schwann cells, which are indispensable for tissue formation and regeneration, 7 days after surgery [41]. These findings indicate that the inflammation stage occurs for approximately 3 days after surgery and that tissue formation starts from approximately 7 days after surgery. Our present results reveal that the DCN expression is increased around the wounded PDL tissue on day 5 after surgery. These results suggest that DCN contributes to the healing of periodontal tissue, especially in the transition stage from inflammation to tissue formation.

DCN is expressed in periodontal tissue, including PDL tissue and the alveolar bone [26,27,28]. Furthermore, proinflammatory cytokines such as IL-1β and TNF-α upregulate DCN production [17,18]. Consistent with previous reports, our results demonstrate that the IL-1β and TNF-α treatment of human PDL cells upregulates DCN expression. Furthermore, in our animal model, the DCN expression was higher in the wounded PDL tissue 5 days after surgery compared with that in the nonwounded tissues. PDL tissue contains a variety of cell populations including PDLSCs that contribute to periodontal tissue healing and regeneration [42,43,44]. These results indicate that DCN produced by proinflammatory cytokines in PDL cells may contribute to the regulation of inflammation and tissue formation in the wounded periodontal tissue, including PDL tissue and the alveolar bone.

To investigate the contribution of DCN to tissue formation in wounded PDL tissue and the alveolar bone, we first focused on the effects of DCN on the osteoblastic differentiation of HPDLSCs and preosteoblasts. DCN plays an important role in osteogenesis [19] and promotes the proliferation and mineralization of osteoblasts [21]. Additionally, DCN is produced by osteoblasts [45], and its expression increases during bone formation [46,47]. Consistent with these reports, we found that DCN promotes the osteoblastic differentiation of HPDLSCs and preosteoblasts. Furthermore, our results show that the upregulation of mineralization and the expression of bone-related genes by DCN are more remarkable in HPDLSCs than in preosteoblasts. These results suggest that DCN mainly promotes the osteoblastic differentiation of HPDLSCs rather than preosteoblasts in periodontal tissue.

We next analyzed the intracellular signaling of the DCN-induced osteoblastic differentiation in HPDLSCs. Previous reports have shown that the ERK and AKT pathways are important for the osteoblastic differentiation of HPDLSCs [48]. Furthermore, DCN signaling is mediated through the ERK and AKT pathways in various cell types [35,36,37]. However, our results show that the phosphorylation of ERK1/2, but not AKT, is upregulated during the DCN-induced osteoblastic differentiation of HPDLSCs. This discrepancy might be because of differences in the cell types. Thus, the DCN treatment of HPDLSCs induces osteoblastic differentiation through the ERK1/2 pathway.

Successful periodontal regeneration results in the formation of new alveolar bone and PDL tissue. For the regeneration of periodontal tissue, PDLSCs need to migrate to the defect site, proliferate, and differentiate into the appropriate cell types [49]. DCN is known to promote the proliferation and migration of human keratinocytes [50], whereas it serves as a negative regulator of the proliferation and migration of trophoblasts [51] and endothelial cells [52]. In this study, we analyzed the effects of DCN on the proliferation and migration of HPDLSCs and showed that DCN does not affect these functions in HPDLSCs. Further studies are needed to determine the mechanism of proliferation and migration of HPDLSCs.

In our current study, DCN does not affect the PDL-related gene expression in HPDLSCs. This suggests that DCN produced in PDL cells under inflammatory conditions functions in the osteoblastic differentiation of HPDLSCs and not in fibroblastic differentiation. Additionally, our immunohistochemical data revealed that the DCN expression is not increased in the osteoblastic layers around the wounded alveolar bone. Furthermore, DCN does not induce the osteoblastic differentiation of preosteoblasts in vitro. Taken together, DCN may contribute to bone healing in wounded periodontal tissue by the induction of osteoblastic differentiation in HPDLSCs but not in preosteoblasts.

## 4. Materials and Methods

### 4.1. Cell Culture

Human PDL cells were isolated from the teeth of three healthy individuals who visited Kyushu University for extraction: a third molar from a 23-year-old male (3S), a premolar from a 15-year-old female (5I), a premolar from a 21-year-old female (5J), a premolar from a 17-year-old female (5L), and a premolar from a 25-year-old female (5T) as described previously [32]. Briefly, the middle region of the periodontal ligament was stripped from the root surface of the extracted tooth. The obtained tissue was cut into pieces of about 0.5 cm^2^ and washed with alpha minimum essential medium (α-MEM; Gibco-BRL, Grand Island, NY, USA) supplemented with 50 μg/mL of streptomycin and 50 U/mL of penicillin. Then, the tissue was incubated at 37 °C for 20 min in α-MEM containing 0.2% collagenase (Wako Pure Chemical Industries Ltd., Osaka, Japan) and 0.25% trypsin (Nacalai Tesque, Kyoto, Japan), and the cells were dispersed. The cells were centrifuged at 1000 rpm for 5 min, and the pellet was suspended in α-MEM containing 10% fetal bovine serum (FBS; Sigma-Aldrich, St. Louis, MO, USA) and seeded onto 35-mm Primaria dishes (Becton Dickinson Labware, Lincoln Park, NJ, USA). The cells were used as human PDL cells in this study from passages three through five. A cell line of heterogeneous immortalized human PDL cells (2-23), which has been reported previously [53], was used as the HPDLSCs. Saos2 cells (RIKEN, Saitama, Japan) were used as human preosteoblasts. These cells were cultured in α-MEM containing 10% FBS, 50 µg/mL of streptomycin, and 50 U/mL of penicillin (10% FBS/α-MEM). The cells were cultured at 37 °C in a humidified atmosphere with 5% CO_2_. All procedures were performed in compliance with the requirements of the Kyushu University Certified Institutional Review Board for Clinical Trials (approval code: 2-115).

### 4.2. Decorin (DCN) Coating

DCN coating was performed as previously reported [54]. Recombinant human DCN (Sino Biological, Beijing, China) was used to coat 24-well plates at concentrations of 0.5 and 2 μg/mL. The plates were incubated at 37 °C for 8 h, followed by washing twice with phosphate-buffered saline (PBS).

### 4.3. Animal Model

Twenty-three Sprague–Dawley rats (5-week-old males) were purchased (Kyudo, Saga, Japan). Two rats were kept in one cage during breeding. A rat with an infection after surgery and A rat that died during anesthesia were excluded from this study. During the surgery, the body weights of the rats were 140 ± 20 g (mean ± SEM). Based on the previous study [41], animals were anesthetized with an injection of 2.5 mg/kg of butorphanol tartrate (Meiji Seika Pharma Co. Ltd., Tokyo, Japan), 0.15 mg/kg of medetomidine hydrochloride (Kyoritsu Seiyaku Co. Ltd., Tokyo, Japan), and 2 mg/kg of midazolam (Sandoz Inc, Tokyo, Japan) in the animal surgery room. A periodontal defect (2 mm in diameter and depth) was created at the mesiopalatal submarginal portion to the palatal root of the left maxillary first molar. Three rats without surgery were used as controls. At 1, 3, 5, 7, 14, and 28 days post-surgery, animals were intracardially perfused with 4% paraformaldehyde (Merck Millipore, Darmstadt, Germany) in PBS. Samples were resected and decalcified in Kalkitox (Wako) at 4 °C for 48 h, followed by neutralization using a 5% sodium sulfate solution (Wako) for 24 h. The samples were then embedded in OCT compound (Sakura Finetek, Tokyo, Japan). All procedures were performed under the approval of the Animal Ethics Committee and followed the regulations of Kyushu University (approval code: A22-030-0).

### 4.4. Immunochemical Analysis

Sections were prepared (7-μm-thick) and blocked with 2% bovine serum albumin in PBS for 1 h. Then, the sections were incubated with a rabbit polyclonal anti-DCN antibody (1:100 dilution; Proteintech, Rosemont, IL, USA), a rabbit monoclonal anti-OSX antibody (1:500 dilution; abcam, Cambridge, UK), or normal rabbit IgG (Cell Signaling Technology, Beverly, MA, USA) at 4 °C overnight. The sections were then incubated with a biotinylated goat anti-rabbit secondary antibody, followed by an avidin–peroxidase conjugate (Nichirei Biosciences, Inc., Tokyo, Japan) and DAB solution (Nichirei). Nuclei were counterstained with Mayer’s hematoxylin solution (Wako). The anti-DCN antibody-positive area was measured using a Biozero digital microscope (Keyence Corporation, Osaka, Japan). For the quantification of anti-DCN antibody-positive areas, four regions were selected in each sample of the wounded PDL tissue or osteoblastic layers around the wounded alveolar bone, and the anti-DCN antibody-positive areas in these regions were calculated. The osteoblastic layers were determined by the anti-OSX antibody-positive areas around the wounded alveolar bone (Appendix A). For the control, the anti-DCN antibody-positive areas in the same regions were calculated in the normal periodontal tissue. The values were calculated using the ratio of the two values of the control and the wounded site, which were obtained by calculating the mean value of the four regions.

### 4.5. Treatment of Human PDL Cells with IL-1β and TNF-α

Human PDL cells 3S, 5I, 5J, 5L, and 5T (6 × 10^4^ cells/well) were cultured in six-well plates with 2 mL of culture medium for 24 h and then treated with 10 ng/mL of recombinant human IL-1β (PeproTech EC, London, UK) and 10 ng/mL of recombinant human TNF-α (PeproTech EC) for 6 h. The concentrations of these cytokines were determined in a previous study [55].

### 4.6. Osteoblastic Differentiation of HPDLSCs and Preosteoblasts

Our recent study revealed that calcium supplementation can induce the osteoblastic differentiation and mineralization of HPDLSCs [56] and Saos2 cells [57]. Thus, we used CaCl_2_ for the induction of osteoblastic differentiation of these cells. Furthermore, we confirmed the proliferation and morphology of HPDLSCs and Saos2 cells during osteoblastic differentiation by CaCl_2_ treatment (Appendix A). HPDLSCs and Saos2 cells (1 × 10^4^ cells per well) were cultured in 24-well plates in four different media: 10% FBS/α-MEM as a control medium (CM), CM with 1.5 mM CaCl_2_ (DM), DM on a 0.5 μg/mL DCN coating (DM + 0.5 DCN), and DM on a 2 μg/mL DCN coating (DM + 2 DCN). After 2 weeks of HPDLSC culture and 12 days of Saos2 cell culture, Alizarin red S staining was performed. Alizarin red S-positive areas were measured under a Biozero digital microscope (Keyence). The total RNA was extracted for quantitative reverse transcription-polymerase chain reaction (RT-PCR) analysis after 12 h or 7 days of HPDLSC culture and 12 h or 10 days of Saos2 cell culture.

### 4.7. Quantitative RT-PCR

Total RNA was isolated with TRIzol reagent (Invitrogen, Carlsbad, CA, USA). First-strand cDNA was synthesized from 1 mg of the total RNA using an ExScript RT Reagent kit (Takara Bio Inc., Kusatsu, Japan) in accordance with the manufacturer’s instructions. PCR was performed using KAPA Express Extract (Kapa Biosystems, Woburn, MA) in a Thermal Cycler Dice Real Time System (Takara Bio Inc., Siga, Japan). Primer sequences, annealing temperatures, cycle numbers, and product sizes for *bone morphogenetic protein 2* (*BMP2*), *bone sialoprotein* (*BSP*), *decorin* (*DCN*), *fibrillin 1* (*FBN1*)*, osteocalcin* (*OCN*), *osteoprotegerin* (*OPG*)*, osteopontin* (*OPN*), *osterix* (*OSX*), *periodontal ligament associated protein 1* (*PLAP1*)*, periostin* (*POSTN*), and *β-actin* are shown in Table 1. *β-actin* was used as the internal control. The expression levels of the target genes were calculated using 2^(−ΔΔCt) values.

### 4.8. Western Blotting

HPDLSCs (2 × 10^4^ cells) were seeded into 35 mm dishes and cultured in CM, DM, or DM + 2 DCN for 15 min. The cells were lysed in Pierce RIPA buffer (Thermo Fisher Scientific Inc., Waltham, MA, USA) and were supplemented with 1% protease inhibitor cocktail (Sigma-Aldrich) and 1% phosphatase inhibitor cocktail (Thermo Fisher Scientific Inc.). HPDLSCs (4.5 × 10^4^ cells) or Saos2 cells were seeded into 35 mm dishes and cultured in CM, DM, or DM + 2 DCN for 7 or 10 days. The cells were lysed in Pierce RIPA buffer (Thermo Fisher Scientific Inc.) with 1% protease inhibitor cocktail (Sigma-Aldrich). Approximately 10 mg of protein was separated by 10% sodium dodecyl sulfate (Nacalai Tesque) polyacrylamide gel electrophoresis and was transferred onto an Immuno-Blot PVDF membrane (Bio-Rad Laboratories, CA, USA). The membranes were incubated with a mouse monoclonal anti-β-actin antibody (1:1000; Santa Cruz Biotechnology, Dallas, TX, USA), a rabbit monoclonal anti-p-ERK1/2 antibody (1:1000 dilution; Cell Signaling Technology), a rabbit polyclonal anti-total-ERK1/2 antibody (1:1000 dilution; Cell Signaling Technology), a rabbit monoclonal anti-p-AKT antibody (1:2000 dilution; Cell Signaling Technology), a rabbit monoclonal anti-total-AKT antibody (1:1000 dilution; Cell Signaling Technology), or a rabbit polyclonal anti-OPN antibody (1:2000 dilution; Proteintech). The membranes were then incubated with a biotinylated goat anti-rabbit antibody (Nichirei Biosciences, Inc.) or a rabbit anti-mouse antibody (Nichirei Biosciences Inc.). Finally, positive reactions were visualized using the ECL select western blotting detection system (GE Healthcare, Buckinghamshire, UK).

### 4.9. WST-1 Proliferation Assay

HPDLSCs (3 × 10^3^ cells per well) were seeded into 48-well plates coated with or without DCN (2 μg/mL) in 250 mL of medium and were cultured for 3 days. HPDLSCs (3 × 10^3^ cells per well) or Saos2 cells (3 × 10^3^ cells/well) were seeded into 48-well plates in CM or DM and cultured for 3 days. On days 0, 1, and 3 of culture, a WST-1 Cell Proliferation Assay kit (Millipore Corp., Billerica, MA) was used to analyze the HPDLSC and Saos2 proliferation. The cells were incubated with WST-1 reagent at 37°C for 1 h. The absorbance at 450 nm was measured using an ImmunoMini NJ-2300 (System Instruments Co., Ltd., Tokyo, Japan).

### 4.10. Cell Migration Assay

A cell migration assay was performed using cloning rings (AGC Techno Glass Co., Ltd., Shizuoka, Japan). The rings were placed vertically in each well of a 24-well plate, coated with or without DCN (2 μg/mL). Suspensions of HPDLSCs (4 × 10^4^ cells per well) were added carefully to the outside of each ring in the plates. After the cells had firmly attached to the plates, the rings were removed using tweezers, followed by the addition of 1 mg/mL of mitomycin C (Nacalai Tesque) to stop cell proliferation for 1 h. The cells were then washed gently with PBS three times. Images of each well were obtained after 0 and 48 h of culture, and migrated cells were counted and averaged from three random fields per well.

### 4.11. Cytoskeletal Staining

HPDLSCs (2 × 10^4^ cells) and preosteoblasts (2 × 10^4^ cells) were seeded into 35 mm dishes and cultured in CM or DM for 24 h. Cells were fixed with 4% paraformaldehyde (Merck Millipore) in PBS for 15 min, followed by permeabilization with 0.3% Triton X-100 in PBS for 10 min. Then, the cells were stained with Alexa Fluor 488 Phalloidin (Cell Signaling Technology) for cytoskeletal staining. The staining of nuclei was performed using 4,6-diamidino-2-phenylindole (DAPI, Nacalai Tesque). Imaging was performed using a Biozero digital microscope (Keyence Corporation).

### 4.12. Statistical Analysis

All values are expressed as the mean  ±  standard deviation (SD). Statistical analysis was performed using one-way ANOVA, followed by the Bonferroni method for comparisons of three or more groups. Statistical significance was determined as a probability value of *p* < 0.05.

## 5. Conclusions

We found that DCN is an important molecule for the healing of wounded periodontal tissue. The expression of DCN is upregulated around wounded PDL tissue, especially in the early stages of tissue healing, and DCN is produced in PDL cells by proinflammatory cytokines. On the other hand, the DCN expression is not altered in the osteoblastic layers around the wounded alveolar bone. DCN, which is produced under inflammatory conditions, promotes osteoblastic differentiation through ERK1/2 signaling in PDLSCs but not preosteoblasts. Furthermore, in PDL tissue, DCN may not regulate the proliferation, migration, and fibroblastic differentiation of PDLSCs. This study demonstrates that DCN may be a new target for the development of a novel periodontal regenerative therapy. Further in vivo studies are needed to clarify the effects of DCN on the regeneration of periodontal tissue.

## Figures and Tables

**Figure 1 molecules-27-08224-f001:**
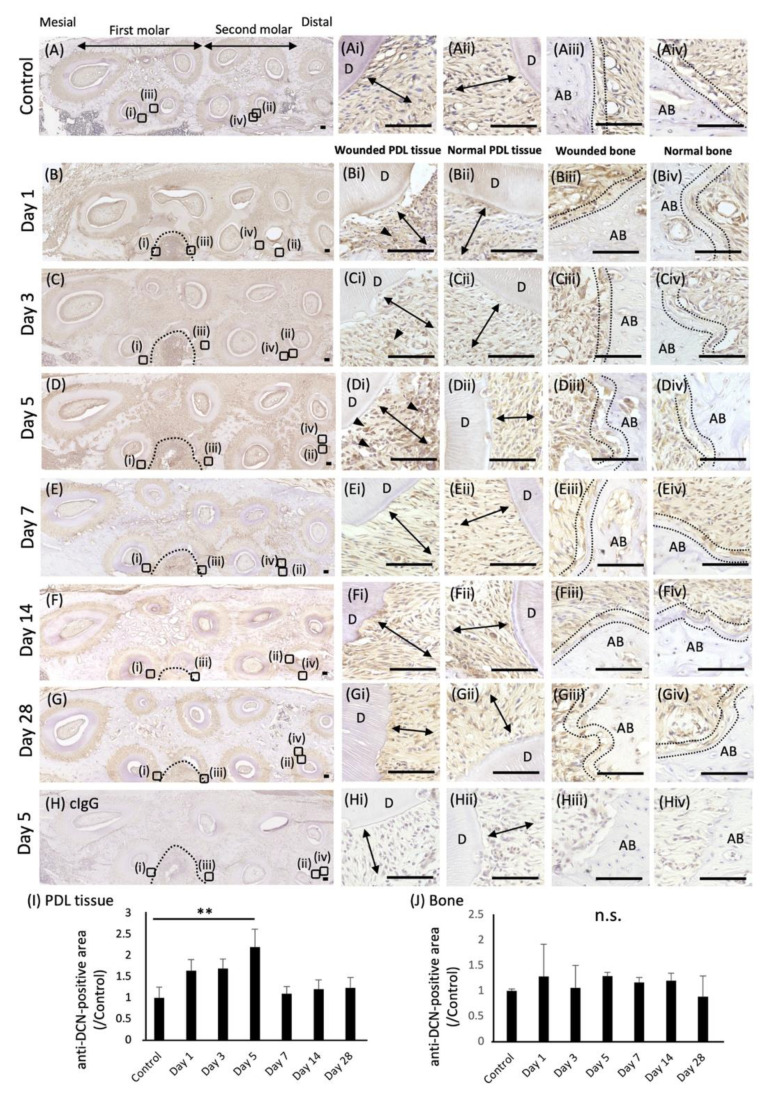
Expression of DCN in wounded rat periodontal tissue. Immunochemical analysis of DCN expression was performed in wounded rat periodontal tissue, including PDL tissue and the alveolar bone. Horizontal sections of rat maxilla were prepared, and preoperative sections were used as a control (**A**). Localization of anti-DCN antibody-positive areas in the wound site was examined on days 1 (**B**), 3 (**C**), 5 (**D**), 7 (**E**), 14 (**F**), and 28 (**G**). Magnified images of wounded PDL tissue (i), normal PDL tissue (ii), wounded alveolar bone (iii), and normal alveolar bone (iv) in each section are shown. Control IgG was used for the negative control (**H**). Nuclei were counterstained with hematoxylin. Arrowheads indicate anti-DCN antibody-positive areas. Double arrows indicate PDL tissue. The inside of the two dotted lines exhibits osteoblastic layers. D, dentin; AB, alveolar bone. Bars = 100 µm. (**I**,**J**) Graphs show the quantification of anti-DCN antibody-positive areas in the PDL tissue (**I**) or the osteoblastic layer around the alveolar bone (**J**). Anti-DCN antibody-positive areas in preoperative periodontal tissue were used as the control. Values are shown as the fold increase relative to the control. ** *p* < 0.01, n.s. = no significance, *n* = 3.

**Figure 2 molecules-27-08224-f002:**
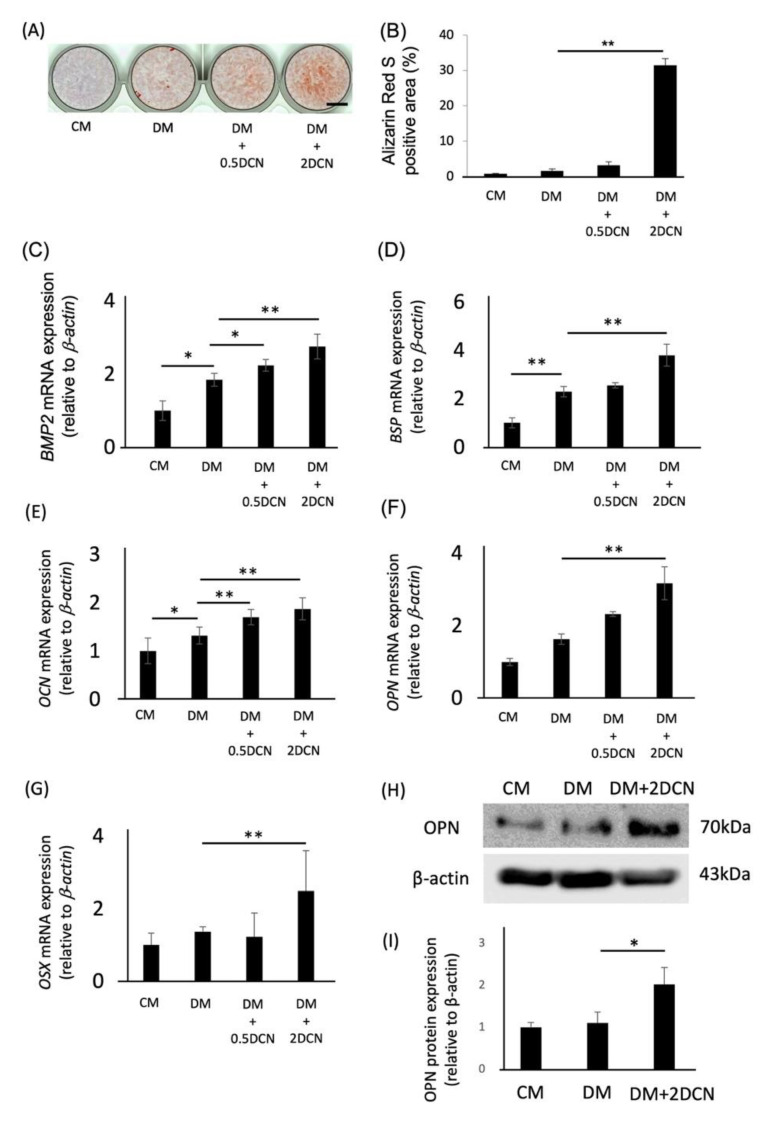
Effects of DCN on the osteoblastic differentiation of HPDLSCs. (**A**) HPDLSCs were cultured in 10% FBS/α-MEM (CM), CM with 1.5 mM CaCl_2_ (DM), DM on a 0.5 μg/mL DCN coating (DM + 0.5 DCN), or DM on a 2 μg/mL DCN coating (DM + 2 DCN) for 2 weeks. Alizarin red S staining was performed to evaluate the mineralization of HPDLSCs. Bar = 500 µm. (**B**) The graph shows the quantification of Alizarin red S-positive areas. (**C**–**G**) Quantitative RT-PCR was performed to analyze the gene expression of *BMP2* (**C**), *BSP* (**D**), *OCN* €, *OPN* (**F**), and *OSX* (**G**) in HPDLSCs cultured under the same conditions for 7 days (**C**–**F**) or 12 h (**G**). (**H**,**I**) HPDLSCs were cultured in CM, DM, or DM + 2 DCN for 7 days. (**H**) Western blot analysis was performed to determine the expression of OPN. β-actin was used as a loading control. (**I**) The graph shows the quantification of OPN expression. Normalization of protein expression was performed against β-actin expression. Values are the mean ± SD of three indepenDent. experiments. ** *p*  <  0.01, * *p*  <  0.05.

**Figure 3 molecules-27-08224-f003:**
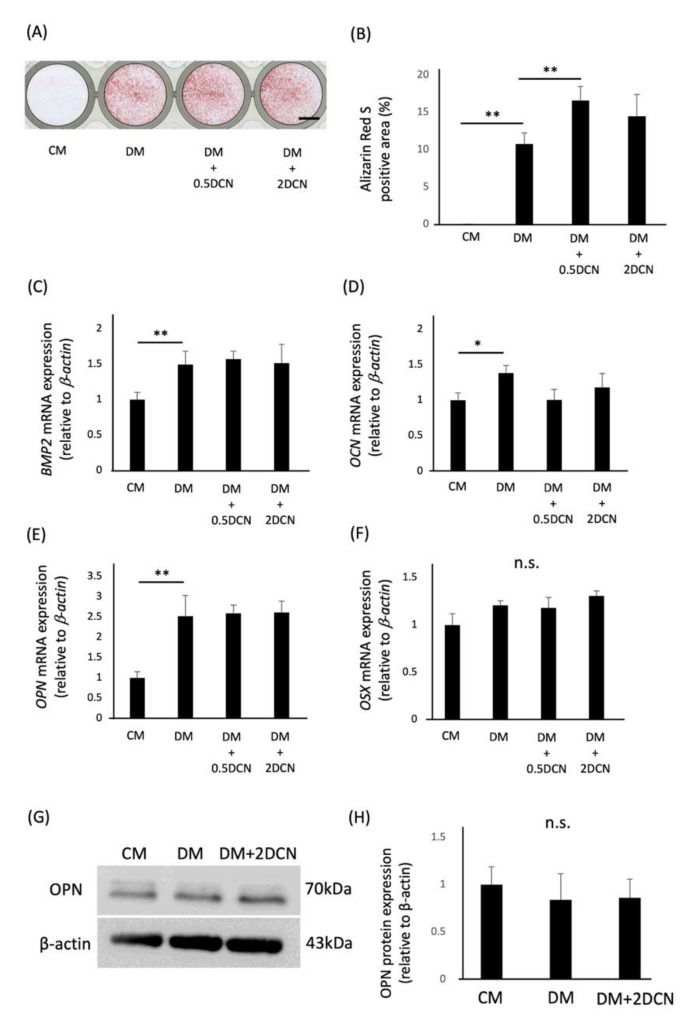
Effects of DCN on the osteoblastic differentiation of preosteoblasts. (**A**) Preosteoblasts were cultured in 10% FBS/α-MEM (CM), CM with 1.5 mM CaCl_2_ (DM), DM on a 0.5 μg/mL DCN coating (DM + 0.5 DCN), or DM on a 2 μg/mL DCN coating (DM + 2 DCN) for 12 days. Alizarin red S staining was performed to evaluate the mineralization of preosteoblasts. Bar = 500 µm. (**B**) The graph shows the quantification of Alizarin red S-positive areas. (**C**–**F**) Quantitative RT-PCR was performed to analyze the gene expression of *BMP2* (**C**), *OCN* (**D**), *OPN* (**E**), and *OSX* (**F**) in preosteoblasts cultured under the same conditions for 10 days (**C**–**E**) or 12 h (**F**). (**G**) Western blot analysis was performed to determine the expression of OPN. β-actin was used as a loading control. (**H**) The graph shows the quantification of OPN expression. Normalization of protein expression was performed against β-actin expression. Values are the mean ± SD of three indepenDent. experiments. ** *p*  <  0.01, * *p*  <  0.05, n.s. = no significance.

**Figure 4 molecules-27-08224-f004:**
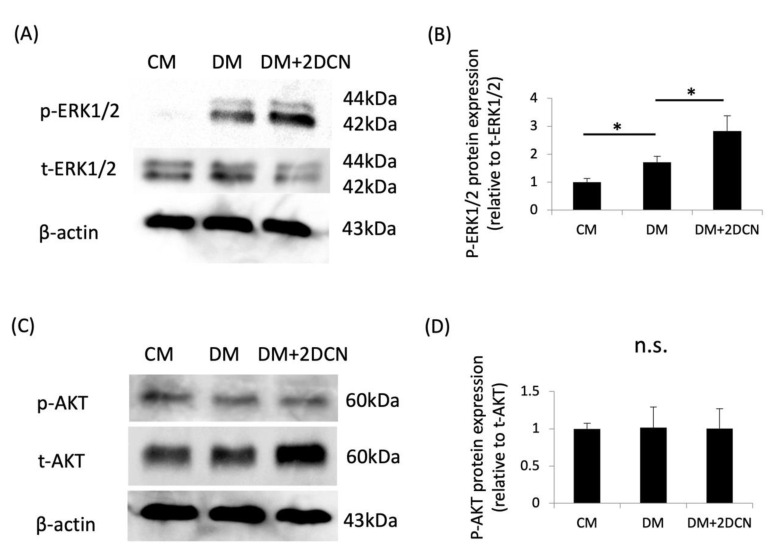
Expression of phosphorylated-ERK1/2 and phosphorylated-AKT during the DCN-induced osteoblastic differentiation of HPDLSCs. HPDLSCs were cultured in 10% FBS/α-MEM (CM), CM with 1.5 mM CaCl_2_ (DM), or DM on a 2 μg/mL DCN coating (DM + 2 DCN) for 15 min. (**A**) Western blot analysis was performed to determine the expression of phosphorylated-ERK1/2 (p-ERK 1/2) and total-ERK1/2 (t-ERK 1/2). β-actin was used as a loading control. (**B**) The graph shows the quantification of p-ERK 1/2 expression. Normalization of protein expression was performed against t-ERK1/2 expression. (**C**) Western blot analysis was performed to determine the expression of phosphorylated-AKT (p-AKT) and total-AKT(t-AKT). β-actin was used as a loading control. (**D**) The graphs show the quantification of p-AKT expression. Normalization of protein expression was performed against t-AKT expression. Values are the mean ± SD of three indepenDent. experiments. * *p* < 0.05, n.s. = no significance.

**Figure 5 molecules-27-08224-f005:**
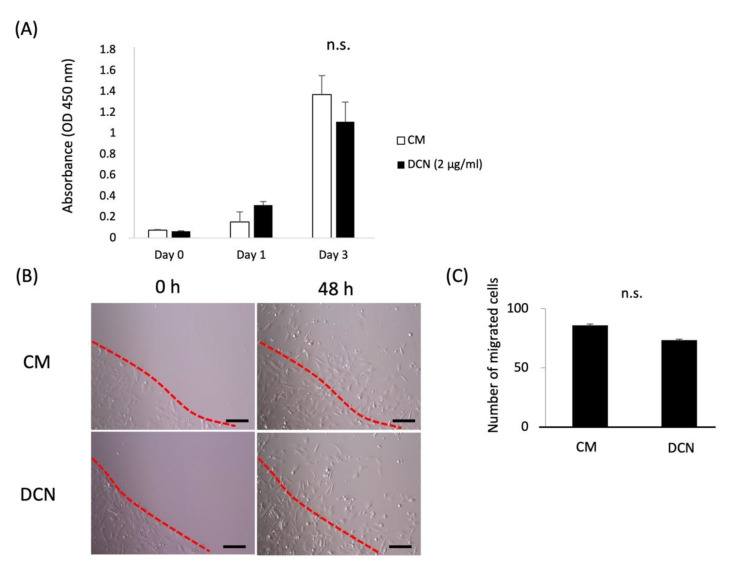
Effects of DCN on the proliferation and migration of HPDLSCs. (**A**) HPDLSCs were cultured in 10% FBS/α-MEM (CM) or CM on a 2 μg/mL DCN coating (DCN) for 0, 1, and 3 days. A proliferation assay was performed using the WST-1 proliferation assay kit at an absorbance of 450 nm. (**B**) Migration of HPDLSCs was analyzed by a ring cell migration assay. HPDLSCs were cultured in CM or DCN for 48 h, and then migrated cells were counted. Dotted lines delineate the ring edges. Bars = 200 µm (**C**) The graph shows the quantification of migrated cells. Values are the average of three random fields per well. n.s. = no significance.

**Figure 6 molecules-27-08224-f006:**
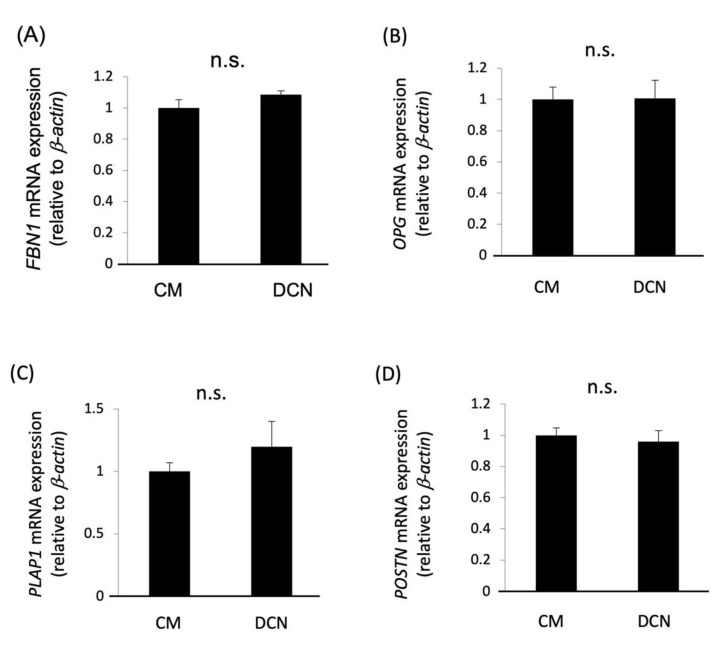
Effects of DCN on the expression of PDL-related genes in HPDLSCs. (**A**–**D**) Quantitative RT-PCR was performed to analyze the gene expression of *FBN1* (**A**), *OPG* (**B**), *PLAP1* (**C**), and *POSTN* (**D**) in HPDLSCs cultured in 10% FBS/α-MEM (CM) or CM on a 2 μg/mL DCN coating (DCN) for 7 days. Values are the mean ± SD of three indepenDent. experiments. n.s. = no significance.

**Table 1 molecules-27-08224-t001:** Primer sequence, product size, annealing temperature, for quantitative reverse transcription polymerase chain reaction.

Target Gene (Abbreviation)	Gene Bank ID	Forward (Top) and Reverse (Bottom) Primer Sequences	Size of Amplified Products (bp)	Annealing Temperture (°C)
BSP	NM_004967.3	5′-CTGGCACAGGGTATACAGGGTTAG-3′ 5′-ACTGGTGCCGTTTATGCCTTG-3′	181	60
BMP2	NM_001200.2	5′-TCCACTAATCATGCCATTGTTCAGA-3′ 5′-GGGACACAGCATGCCTTAGGA-3′	73	60
COL1	NM_000088.3	5′-CCCGGGTTTCAGAGACAACTTC-3′ 5′-TCCACATGCTTTATTCCAGCAATC-3′	148	60
DCN	NM_001920.5	5′-AATCCGGCTGAGAGCTCAGG-3′ 5′-GGGAAGGAGGAAGACCTTGA-3′	118	60
FBN1	NM_000138.5	5′-TGCACCTATGGAACCATGTGATAGA-3′ 5′-AAGTGATCCACTGTGTGCCAACTC-3′	189	60
OCN	NM_199173.4	5′-CCCAGGCGCTACCTGTATCAA-3′ 5′-GGTCAGCCAACTCGTCACCAGTC-3′	112	60
OPG	NM_002546.4	5′-CTCGAAGGTGAGGTTAGCATGTC-3′ 5′-TGGCACCAAAGTAAACGCAGAG-3′	196	60
OPN	NM_001040058.2	5′-ACACATATGATGGCCGAGGTGA-3′ 5′-TGTGAGGTGATGTCCTCGTCTGT-3′	115	60
OSX	NM_001300837.2	5′-GCCATTCTGGGCTTGGGTATC-3′ 5′-GAAGCCGGAGTGCAGGTATCA-3′	129	60
PLAP1	NM_017680.5	5′-ATGGGAGTCTTGCTAACATACCAC-3′ 5′-CAGAAGTCATTTACTCCCACTCTTG-3′	154	60
POSTN	NM_006475.2	5′-CATTGATGGAGTGCCTGTGGA-3′ 5′-CAATGAATTTGGTGACCTTGGTG-3′	167	60
b-actin	NM_001101.5	5′-ATTGCCGACAGGATGCAGA-3′ 5′-GAGTACTTGCGCTCAGGAGGA-3′	89	60

## Data Availability

Not applicable.

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
