# Peer review of "Decorin Promotes Osteoblastic Differentiation of Human Periodontal Ligament Stem Cells"

_molecules, 2022, doi:10.3390/molecules27238224_

Round 1

Reviewer 1 Report

The work submitted for review is interesting and deals with an important problem in dentistry, namely tissue regeneration processes. The content presented is a logical whole and the work is well prepared. However, I have the following questions:

1. Why material from only three patients was used in the paper. Please justify in your paper why this amount is sufficient. 

2. How were the isolated cells characterised. Please provide photographs of the culture obtained in the isolation process as a supplement from each patient. 

3. Do you see any practical applications for the results, e.g. implant coating or other applications. 

4. In my opinion, the conclusions should be expanded and placed in a separate chapter. 

Reviewer 2 Report

The present manuscript characterizes the potential influence of decorin on the osteogenic differentiation process of periodontal ligament stem cells, upon in vivo and in vitro characterization. Despite the potential relevance of the manuscript, several issues need to be revised to increase the quality of the manuscript. The following issues should be addressed:

-        The introduction section is too short, particularly for a more generalist audience. Authors should add a contextualization of the relevance of periodontal tissues in human physiology, approach briefly the major diseases of the periodontium, and the potential relevance of precursor populations (stem cells) on periodontal healing/regeneration. Previous therapeutic approaches with decorin (DCN) on regenerative medicine approaches should also be highlighted.

-        Regarding the expression of DCN in the rat periodontal tissues, quantitative results should be presented not only combined, but also separated within the 4 quantified areas, and adequately discussed – wounded vs non-wounded areas.

-        The characterization of osteoblastic cultures needs to include further parameters such as metabolic activity and cell proliferation. Morphological analysis of both osteoblastic and HPDLSCs needs to be presented, particularly upon cytoskeletal staining.

-        A more complete gene expression assessment, further including relevant transcription factors of the osteogenic programs and, as well, relevant downstream targets needs to be presented.

-        Supplemental figure data should be moved into the manuscript.

-        Figure 6 is excessively speculative and should not be presented.

-        Validation of the attained gene expression data should be confirmed through protein expression.

-        Discussion section should be improved, further comparing the attained data with the current state of the art and suggesting possible hypothesis to justify the attained results.

-        Data regarding the expression of DCN in human PDL cells and preosteoblasts treated with IL-1 and TNF-alfa is not new, it is already established into the literature and does not greatly contribute to the manuscript’s rationale. It should be removed.

-        The protocol description regarding the animal experimental procedure is not adequately detailed. The authors are advised to refer to the ARRIVE guidelines (www.nc3rs.org.uk/arrive-guidelines) to adequately present data on the animal protocol.

-        The manuscript should be carefully checked for abundant language and grammatical errors, preferably by a native English speaker, to correct the attained inconsistencies.

Round 2

Reviewer 2 Report

The authors significantly improved the quality of the manuscript. 

My final recommendation is that data on Fig 5A and Supplemental Fig 2A and B are presented as bar graphs and not line graphs, as authors only evaluated independent time points and do not continuously monitor data.

Author Response

Point 1: My final recommendation is that data on Fig 5A and Supplemental Fig 2A and B are presented as bar graphs and not line graphs, as authors only evaluated independent time points and do not continuously monitor data. 

Response 1: Thank you very much for your suggestion. We changed the data on Fig 5A and Supplemental Fig 2A and B to the bar graphs.